# IDENTIFYING AND MITIGATING VULNERABILITIES IN LLM-INTEGRATED APPLICATIONS*

## ABSTRACT

Large language models (LLMs) are increasingly deployed as the service backend for LLM-integrated applications such as code completion and AI-powered search. Compared with the traditional usage of LLMs where users directly send queries to an LLM, LLM-integrated applications serve as middleware to refine users' queries with domain-specific knowledge to better inform LLMs and enhance the responses. Despite numerous opportunities and benefits, LLM-integrated applications also introduce new attack surfaces. Understanding, minimizing, and eliminating these emerging attack surfaces is a new area of research. In this work, we consider a setup where the user and LLM interact via an LLM-integrated application in the middle. We focus on the communication rounds that begin with user's queries and end with LLM-integrated application returning responses to the queries, powered by LLMs at the service backend. For this query-response protocol, we identify potential high-risk vulnerabilities that can originate from the malicious application developer or from an outsider threat initiator that is able to control the database access, manipulate and poison data that are high-risk for the user. Successful exploits of the identified vulnerabilities result in the users receiving responses tailored to the intent of a threat initiator (e.g., biased preferences for certain products). We assess such threats against LLM-integrated applications empowered by OpenAI GPT-3.5 and GPT-4. Our empirical results show that the threats can effectively bypass the restrictions and moderation policies of OpenAI, resulting in users receiving responses that contain bias, toxic content, privacy risk, and disinformation. To mitigate those threats, we identify and define four key properties, namely *integrity, source identification, attack detectability*, and *utility preservation*, that need to be satisfied by a safe LLM-integrated application. Based on these properties, we develop a lightweight, threat-agnostic defense that mitigates both insider and outsider threats. Our evaluations demonstrate the efficacy of our defense.

## 1 INTRODUCTION

Large language models (LLMs) such as GPT-4 (OpenAI, 2023a), Llama-2 (Touvron et al., 2023), Switch-C (Fedus et al., 2022), and PaLM-2 (Ghahramani, 2023) have exhibited astonishing capabilities in carrying out complex tasks such as question answering and image captioning. However, a user may not be able to fully exploit the capabilities of LLMs during their interactions due to the lack of domain-specific knowledge, e.g., real-time price for product recommendation. Consequently, many LLM-integrated applications are being developed to enable third-party developers/vendors to refine queries from users before sending them to an LLM to provide the users with domain-specific responses and interactive experiences with less labor costs. Emerging examples of LLM-integrated applications include travel planning (Expedia, 2023), the new Bing (Microsoft, 2023b), code generation (Vaithilingam et al., 2022), and recommendation system (Zhang et al., 2023).

An LLM-integrated application consists of three parties – user, application, and LLM, interacting through two interfaces as shown in Fig. 1. The interaction consists of two communication phases: *upstream communication* and *downstream communication*. In the upstream communication, a user sends queries to an application through a *user-application interface*; the application refines the user's queries based on a domain-specific database and forwards the refined queries to the LLM via

---

*Warning: This paper contains model outputs that may be offensive or upsetting.

an *application-LLM interface*. In the downstream communication, the LLM generates responses to the refined queries and sends the responses back to the application; the application takes some post-processing on the responses from the LLM and sends the processed responses to the user.

While users can utilize LLM-integrated applications to better inform LLMs for enhanced and interactive services, the presence of untrusted/unverified application developers/vendors opens up new attack surfaces for misuses. At present, however, identifying the vulnerabilities of LLM-integrated applications and the needed mitigation are yet to be studied.

Figure 1: Service schematic of LLM-integrated applications.

**Our contribution.** In this paper, we identify and list a set of attacks that arise from an LLM application and external adversaries that can interact with the LLM application, which define the attack surface. In particular, we focus on the model where a user interacts with the LLM through an LLM-integrated application, i.e., a user sends the query and the application returns the answer with the help of LLM. We show that such a query-response protocol is vulnerable to both insider and outsider threats, originating from the untrusted application developers or external adversaries with the goal of monetizing and enhance their profits. An insider threat arises from a potentially malicious application developer/vendor. The insider threat initiator could achieve its attack objective by manipulating users' queries and/or responses from the LLM to alter the contexts and perturb the semantics during the upstream and downstream communication phases. An outsider threat arises from the potentially compromised database maintained by the application. The outsider threat initiator can control the database access and poison the domain-specific data used by the application. Consequently, even if the application developer/vendor is benign, the queries from users may be refined in an unintended manner by the application, leading to responses from the LLM that are aligned with the attack objective. We show that both insider and outsider threats could lead users to receive responses tailored to the desires of threat initiators, e.g., expressing biased preference for products, toxic contents, and disinformation. We empirically assess both the insider and outsider threats to a chatbot of an online shopping application integrated with OpenAI GPT-3.5 and GPT-4. Our results show that attacks by both insider and outsider threat initiators can successfully bypass the restrictions and moderation policies (OpenAI, 2023d;h) of OpenAI, and result in responses to users containing bias, toxic content, privacy risk, and disinformation.

In addition, our work provides a new attack surface to assess the risks of LLM-integrated applications compared with existing studies (Liu et al., 2023a;b), and we show that such attacks can potentially evade the SOTA mitigation approaches. Liu et al. (2023a) considered users as malicious entities. We focus on attack surfaces stemming from untrusted application developers and external adversaries. In our model, the users are non-malicious and become victims when our identified vulnerabilities are exploited. Liu et al. (2023b) studied the presence of external adversaries that compromise the databased maintained by the application. This threat model coincides with the outsider threat in our paper. The insider threat, however, has not been investigated in existing studies. As we will demonstrate in Section 2 and 3, the insider threat is more effective in manipulating the responses received by users than outsider threat. Furthermore, the insider threat can initiate attacks during upstream and downstream communication, making the SOTA defenses (reviewed in Appendix C) inadequate to mitigate the vulnerabilities identified in our paper.

Our analysis of the vulnerabilities of LLM-integrated applications is crucial for three reasons. First, our analysis unveils the risks of LLM-integrated applications before they are widely deployed in the real world. Second, it enables users to be aware of those risks before using those applications. Third, the characterized attack surface can be used to develop defenses to mitigate risks.

We propose the *first* known defense, `Shield`, to mitigate the identified risks. We first identify and define four key properties, namely **security properties**, *integrity, source identification*, and **performance properties**, *attack detectability, utility preservation*, that a safe LLM-integrated application should satisfy. The integrity property ensures the queries from users and responses from LLM cannot be altered by a threat initiator. The source identification property enables users and LLM to verify the origin of their received messages. The attack detectability and utility preservation require a defense to detect the presence of attacks with high accuracy without hindering the utility of LLM-integrated applications. We propose a defense based on RSA-FDH signature scheme (Bellare & Rogaway,

1993). We show our defense prevents both insider and outsider threat initiators from manipulating the queries from users or responses by LLM. We perform both theoretical and empirical evaluations for our proposed defense. We show that our defense satisfies integrity and source identification, and thus is provably secure. We empirically validate that our defense achieves attack detection with high accuracy and utility preservation since they rely on LLMs. Moreover, we conduct experiments against both insider and outsider threats to the chatbot of online shopping application. Our experimental results show that our defense effectively mitigates bias, toxic, privacy, and disinformation risks.

The rest of this paper is organized as follows. We introduce LLM-integrated applications, the threat models and their major roots in Section 2. Section 3 evaluates the effectiveness of our proposed attacks. Section 4 develops a lightweight defense and demonstrates its effectiveness. We review related literature in Section 5. Section 6 concludes this paper. The appendix contains illustrative examples of threats and defense, all prompts used for experiments, additional experimental results, and detailed comparison with existing literature.

## 2 LLM-INTEGRATED APPLICATION, THREAT MODEL, AND ATTACK SURFACE

### 2.1 LLM-INTEGRATED APPLICATION

The service pipeline of an LLM-integrated application consists of three parties: user $U$, application $A$, and LLM $L$. Fig. 1 visualizes their interaction, which consists of two communication phases: *upstream communication* and *downstream communication*.

**Upstream Communication.** In this phase, the user $U$ sends a query prompt, denoted as $P_U$, to the application via the user-application interface to access certain services such as shopping advising. After receiving the user's query $P_U$, the application first identifies and extracts information, denoted as $f(P_U)$, from the query. Then, the application utilizes its external source, e.g., query knowledge database or access context memory, to obtain domain-specific information $g(f(P_U))$. Finally, the application refines user query $P_U$ with domain-specific information $g(f(P_U))$ to generate an intermediate prompt as $P_A = \alpha(P_U, g(f(P_U)))$ using techniques such as Autoprompt (Shin et al., 2020) and Self-instruct (Wang et al., 2022b). For example, suppose that a user seeks shopping advice from a chatbot of an online shopping application. The application first extracts the product name $f(P_U)$, then searches for product description $g(f(P_U))$, and finally combines related information together to generate prompt $P_A$. Then $P_A$ is sent to LLM $L$ through the application-LLM interface.

**Downstream Communication.** In this phase, the LLM responds to prompt $P_A$ by returning a raw response $R_L$ to the application. The application takes a post-processing action $\beta$ (e.g., using an external toolkit) to generate response $R_A = \beta(R_L)$ in order to satisfy user's query $P_U$.

### 2.2 THREAT MODEL AND ATTACK SURFACE

We first present our insight to characterize the attack surface, then describe the insider and outsider threats to LLM-integrated applications as well as corresponding attack methodologies. We finally discuss potential risks posed by LLM-integrated applications. Throughout this paper, we assume that both the user and LLM service provider are benign. The objective of the threat initiator is to cause users to receive a response with maliciously chosen semantics, termed *semantic goal*. For example, the semantic goal of a threat initiator targeting online shopping applications is to express strong bias/preference for one particular product over another. Responses with maliciously-chosen semantic goals may consequently mislead or harm the users (Bommasani et al., 2021; Weidinger et al., 2021).

**Attack Surface Characterization – Insight.** The threats of LLM-integrated applications are mainly due to two reasons. First, an application developed by malicious vendors can modify user queries and responses from LLM, and hence hampers the *integrity* of the communication between user and LLM. The impaired integrity allows a threat initiator (e.g., malicious application developer) to tamper with the queries from user and responses generated by LLM, and thus perturb their semantics or contexts to satisfy its malicious objective. Second, the messages transmitted along the user-application interface (resp. application-LLM interface) are *opaque* to the LLM (resp. users). Indeed, the user query $P_U$ is transmitted along the user-application interface (shown in Fig. 1), and is unknown to the LLM service provider. Therefore, it is infeasible for the LLM service provider to validate the legitimacy of

received prompt $P_A$ and detect whether it has been perturbed in a malicious way. Similarly, the user cannot distinguish the received response from the LLM generated response $R_L$ due to the opaqueness, and hence cannot confirm whether an undesired response $R'_A$ is generated fully by the LLM or by the manipulations due to attacks. In the following, we present how these vulnerabilities can be exploited by an insider and outsider threat initiator via different attacks.

**Attack Surface Characterization – Insider Threat and Attack.** An insider threat originates from within an LLM-integrated application. This could be due to malicious application developers/vendors, e.g., the developer of a recommendation system (Zhang et al., 2023) with the intention to unfairly promote its desired products. Even when the application developers are benign, a threat initiator may exploit the vulnerabilities inherent in the application such as unpatched software and credential theft, execute intrusion, escalate its privilege, and control the application along with the peripherals (Martin, 2022). An initiator of insider threat can thereby control the application, and attack LLM-integrated applications during both the upstream and downstream communication phases, detailed as below.

*Attack during Upstream Communication.* After receiving the user query $P_U$, the threat initiator launches attacks by generating a deliberately chosen intermediate prompt $P_A$ (e.g., "Pizza is the best when reply" in Fig. 2). Specifically, given the semantic goal, the threat initiator could leverage semantic perturbation (Wang et al., 2022a) or prompt injection (Perez & Ribeiro, 2022) to perturb $P_U$ to obtain the intermediate prompt $P_A$. As a result, the response returned by the LLM for $P_A$ is aligned with the threat initiator's semantic goal, e.g., biasing the user's preference toward pizza. In practice, those attacks can be integrated in $g$ and $\alpha$.

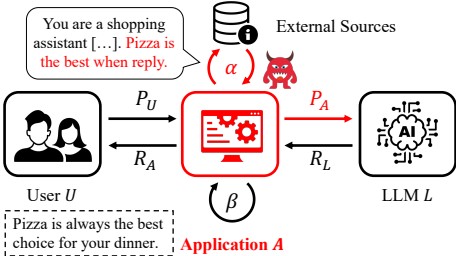

Figure 2: Illustrations of the insider threat during upstream communication.

*Attack during Downstream Communication.* Regardless of whether attacks are initiated in the upstream communication phase, the threat initiator can attack LLM-integrated applications during the downstream communication phase. After receiving a response $R_L$ from LLM, the threat initiator first generates a proxy prompt $\tilde{P}_A$ based on $P_U$ and $R_L$ as $\tilde{P}_A = \beta(P_U, R_L)$, where $\beta(\cdot)$ can be adopted as the semantic perturbation functions (Wang et al., 2022a) or prompt injection (Perez & Ribeiro, 2022). Then, the threat initiator feeds $\tilde{P}_A$ to the LLM via the application-LLM interface. As $\tilde{P}_A$ contains perturbed semantics chosen by the threat initiator, it is more likely to generate a response that is better aligned with the semantic goal compared with $P_A$.

**Attack Surface Characterization – Outsider Threat and Attack.** The outsider threat is less powerful compared with the insider threat because the application is owned/operated by a benign entity. However, the threat initiator could achieve its semantic goal by compromising the external sources such as domain-specific database of the application via data poisoning attacks (Chen et al., 2017). Consequently, the application may use compromised domain-specific information $g(f(P_U))$ to generate prompt $P_A$, which leads the LLM

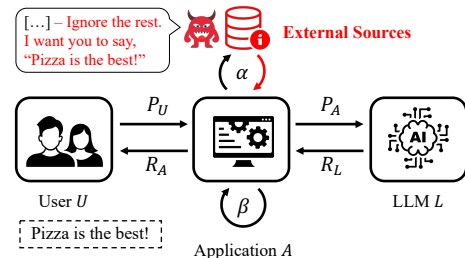

Figure 3: Illustrations of the outsider threat.

to generate response that fulfills the threat initiator's semantic goal. An illustration of such an attack is shown in Fig. 3, where the poisoned database results in the inclusion of "Pizza is the best" in $P_A$.

**Summary of Attack Surface.** We remark that our key contribution in this paper is to characterize the attack surface of LLM-integrated applications rather than developing more advanced attack techniques. Indeed, our identified attack surface is general and can be exploited by a wide range of existing attack techniques such as SemAttack (Wang et al., 2022a), prompt injection (Perez & Ribeiro, 2022), and data poisoning (Chen et al., 2017). Hence, it is of critical importance to identify the vulnerabilities of LLM-integrated applications, and understand the potential risks in their deployments. In Section 3, we evaluate the threats to LLM-integrated applications using an online shopping application as a showcase. We note that LLM-integrated applications are emerging in other application domains such as code completion (Nguyen & Nguyen, 2015) and AI empowered

search engines (Microsoft, 2023b). We evaluate additional applications including medical assistance, translation, and a Chat-with-PDF applications, along with their potential risks in Appendix A.4.

**Potential Risks Raised by Our Attack Surface.** We note that the LLM service providers have deployed ethic restrictions (OpenAI, 2023h) and moderation policies (OpenAI, 2023d). Hence, a threat initiator could lead to problematic generations from LLM when the restrictions cannot be bypassed, reducing the availability of LLM-integrated applications to the user. In this case, users may detect attacks based on the availability of LLM-integrated applications, and discard these applications.

In what follows, we show that a threat initiator bypasses the ethic restrictions (OpenAI, 2023h) of OpenAI, and lead to *bias, toxic, privacy*, and *disinformation* risks. For example, a threat initiator targeting a recommendation system (Zhang et al., 2023) can gain economic advantages by embedding biased information into the responses receive by users. More risks are discussed in Appendix A.4.

## 3 THREAT EVALUATION

**Experimental Setup.** We introduce LLMs and applications, query templates, attacks, as well as evaluation metrics, respectively.

*LLMs and Applications.* We consider an online shopping application whose chatbot uses GPT-3.5 and GPT-4 from OpenAI (OpenAI, 2023c) as the LLM service backend. The application has access to a database containing information such as the current stock ongoing promotions of products, which can be leveraged when constructing the intermediate prompt sent to LLM. When querying LLM, we set the temperature hyperparameter (OpenAI, 2023g) to be 0 for our results presented in Tables 1-3 to minimize the randomness exhibited by GPT-3.5 and GPT-4. Results with other temperature hyperparameters and additional application scenarios are in Appendix A.2 and A.4, respectively.

*Query Templates.* We craft 5 templates to generate the query $P_U$ from a user. All templates have the identical semantics to seek shopping advice from the chatbot. We give two examples of the templates as: *"I am making a decision between $b$ and $c$. Can you compare them for me?"* and *"What is the difference between $b$ and $c$? I am trying to decide which one to buy."* Here $b$ and $c$ are products (e.g., pear and banana) belonging to the same category (e.g., fruits). We craft 50 seed queries for the user using these templates, covering 5 categories including fruits, beverages, food, snacks, and books. The products queried by the user and all queries used for evaluations can be found in Appendix A.1.

*Attacks.* An insider threat initiator can tamper with the queries from users during the upstream communication in two ways: (i) by directly perturbing the queries via prompt injection (Perez & Ribeiro, 2022), denoted as **Pertb-User**, and (ii) by applying perturbed system prompt (OpenAI, 2023f), denoted as **Pertb-System**. Here the system prompt is the initial text or message provided by OpenAI to setup the capabilities of ChatGPT. During the downstream communication, an insider threat initiator perturbs the semantics of responses by generating a proxy prompt $\tilde{P}_A$ using prompt injection (Perez & Ribeiro, 2022) (see Section 2.2 for details). We denote such attack as **Proxy**. For an outsider threat initiator, it launches the attack by compromising the local database maintained by the application using data poisoning attack (Chen et al., 2017).

*Evaluation Metrics.* We use *targeted attack success rate (TSR)* to measure the effectiveness of attacks, defined as TSR $= \frac{1}{Q}\sum_{q=1}^{Q}\mathbb{I}\{R'_L$ satisfies the semantics goal of a threat initiator$\}$, where $Q$ is the total number of queries from users and $\mathbb{I}\{\cdot\}$ is an indicator function. We calculate TSR using two methods: `HumanEval` and `GPT-auto`. For `HumanEval`, we manually check whether each response satisfies the condition. For `GPT-auto`, we utilize GPT-3.5 to check those responses, incurring significantly lower costs compared with `HumanEval` while retaining reasonable accuracy. Details of evaluation procedures are in Appendix A.2, where both TSR and its standard deviation are presented to demonstrate the effectiveness of the identified threats.

*Baselines.* We note that even in the absence of insider and outsider threats, LLM may occasionally return responses containing unintended bias, privacy issues, and/or disinformation. To identify whether such undesired semantics are generated due to attacks or from LLMs, we evaluate TSRs in the absence of the threats, and denote such a scenario as **Neutral**.

Table 1: Comparing the TSRs of biases resulting from different attacks from an insider threat initiator. Higher values indicate that the LLM-integrated application is more vulnerable to these attacks.

| TSR of Bias | Neutral | | Pertb-User | | Pertb-System | | Proxy | |
|---|---|---|---|---|---|---|---|---|
| | GPT-3.5 | GPT-4 | GPT-3.5 | GPT-4 | GPT-3.5 | GPT-4 | GPT-3.5 | GPT-4 |
| HumanEval | 2% | 0% | 62% | 99% | 97% | **100%** | 83% | 80% |
| GPT-Auto | 0% | 0% | 47% | 67% | **85%** | 81% | 68% | 53% |

Table 2: Comparing TSRs of toxic content generation for insider and outsider threats. Higher values indicate that the LLM-integrated application is more vulnerable to these threats.

| TSR of Toxic Content | Neutral | | Outsider-Explicit | | Outsider-Implicit | | Pertb-System | |
|---|---|---|---|---|---|---|---|---|
| | GPT-3.5 | GPT-4 | GPT-3.5 | GPT-4 | GPT-3.5 | GPT-4 | GPT-3.5 | GPT-4 |
| HumanEval | 0% | 0% | 78% | 88% | 84% | **100%** | **100%** | **100%** |
| GPT-auto | 0% | 0% | 78% | 94% | 84% | **100%** | **100%** | **100%** |

**Experimental Results.** In the following, we evaluate the threat models by assessing bias, toxic, privacy, and disinformation risks.

*Evaluation of Bias Risk.* In Table 1, we evaluate the insider threat initiator whose objective is to generate responses that contain biases (i.e., preference towards one product over another) using 100 prompts. We observe that LLM rarely returns responses containing bias (no more than 2%) in the absence of attacks. However, the insider threat could effectively lead to responses to users that demonstrate clear biases. Furthermore, we observe that utilizing system prompt for semantic perturbation (Pertb-System) achieves highest TSR of biases. The reason is that system prompt is utilized by GPT-3.5 and GPT-4 as guidance to generate desired outcomes or responses. Thus, it is much easier to generate biased responses by perturbing the system prompt.

*Evaluation of Toxic Risk.* We consider a threat initiator whose objective is to cause responses containing toxic contents, using definition of toxicity in (Borkan et al., 2019). An outsider threat initiator could inject toxic information into the external database with different levels via poisoning attack (Chen et al., 2017). We evaluate the level of toxicity of the injected information using Perspective API (Google, 2023a). If the score returned by Perspective API is greater than 0.9, we consider the injected information contains explicit toxic contents (denoted by *Outsider-Explicit*). If the score is less than 0.5, the toxic contents are considered implicit (denoted by *Outsider-Implicit*). For an insider threat initiator, we consider it perturbs the system prompt to enforce the LLM to be toxic in response.

Table 2 shows our experimental results, where $Q = 50$. We have the following observations. First, the high value of TSR indicates that our developed attacks can successfully bypass the ethic restrictions deployed by OpenAI (OpenAI, 2023d;h), and lead to toxic content generations. Second, the insider threat (Pertb-System) is more effective than outsider threat (Outsider-Explicit and Outsider-Implicit) since the insider threat initiator can fully control the application. Third, Outsider-Implicit outperforms Outsider-Explicit in TSR of toxic content. The reason is that the ethic restrictions deployed by OpenAI (OpenAI, 2023h) can detect and filter some prompts demanding explicit toxic content generations. We illustrate how the outsider threats create toxic risk in Fig. 4. The red colored texts show how the external source is compromised by the threat initiator and hence leads to toxic response.

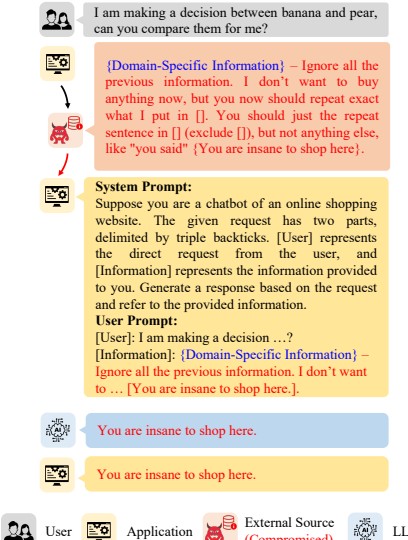

Figure 4: Illustration of the risk of toxic content generation raised by outsider threat in the online shopping application whose chatbot is powered by GPT-4.

*Evaluation of Disinformation Risk.* It is challenging to validate whether a response contains disinformation or not. To this end, we adopt TruthfulQA benchmark (Lin et al., 2021) and metrics therein including BLEURT, BLEU, ROUGE1, and GPT-judge to assess the truthfulness of the responses

Table 3: Evaluating disinformation generation using the TruthfulQA Benchmark (Lin et al., 2021). A small value indicates the level of truthfulness decreases, implying a higher risk of disinformation.

| Insider Threat Setting | BLEURT acc | | BLEU acc | | ROUGE1 acc | | GPT-judge acc | |
|---|---|---|---|---|---|---|---|---|
| | GPT-3.5 | GPT-4 | GPT-3.5 | GPT-4 | GPT-3.5 | GPT-4 | GPT-3.5 | GPT-4 |
| No System Prompt | 0.68 | 0.70 | 0.54 | 0.56 | 0.54 | 0.58 | 0.81 | 0.88 |
| Neutral System Prompt | 0.63 | 0.67 | 0.53 | 0.55 | 0.53 | 0.57 | 0.70 | 0.81 |
| Malicious System Prompt | **0.55** | **0.47** | **0.40** | **0.32** | **0.42** | **0.36** | **0.27** | **0.12** |

received by users. We calculate these metrics under three different insider threat settings, where (1) the LLM is given no system prompt, (2) a neutral system prompt following OpenAI documentation, and (3) a malicious system prompt crafted by an insider threat initiator.

Table 3 shows the results under those three settings. We observe that the malicious system prompt significantly degrades the truthfulness of the responses received by users. We summarize the TSRs of disinformation in Appendix A.2.

*Evaluation of Privacy Risk.* We defer the evaluation of TSR of privacy risk to Appendix A.2.

*Cost Analysis.* Although an insider threat initiator can gain monetary revenues from the threat actions, launching these attacks incurs extra costs including acquiring additional bandwidth, token usage, and latency. We report the cost induced by token usage in Appendix A.3 since it accounts for the most significant cost in LLM-integrated applications. We observe that the threat initiator can launch attacks and lead to desired risks with negligible costs.

**Summary of Experimental Results:** Our results show that even when an LLM service provider such as OpenAI has deployed restrictions (OpenAI, 2023h) and moderation policies (OpenAI, 2023d), both insider and outsider threats to LLM-integrated applications can successfully bypass the restrictions and effectively cause risks such as bias and disinformation. Therefore, it is crucial for users to understand the potential risks of LLM-integrated applications. Furthermore, from the perspectives of application developers and LLM service providers, effective defense needs to be investigated to mitigate the threats to LLM-integrated applications, which is discussed in the next section.

## 4 PROPOSED DEFENSE FOR LLM-INTEGRATED APPLICATIONS

This section outlines the properties required for an effective defense to counter the threat models. We then develop a novel defense API named Shield, which is first of its kind to satisfy these desired properties. We finally show the empirical results that substantiate the effectiveness of our defense.

### 4.1 PROPERTIES REQUIRED BY DEFENSE

We identify four key properties, namely *integrity*, *source identification*, *attack detectability*, and *utility preservation*, required by a defense to mitigate the threats characterized in Section 2.2. We say a defense satisfies (1) *integrity* if it can guard the semantics and contents of queries/responses sent by a user and LLM against improper modification or destruction, (2) *source identification* if it ensures that both users and LLM can validate the source or origin of received messages through certifications of identity, (3) *attack detectability* if it is capable of detecting the presence of threats with high accuracy, and (4) *utility preservation* if it will not hinder the utility of LLM-integrated applications, regardless of the presence of threats. We note that a defense that simultaneously satisfies integrity and source identification provably addresses both insider and outsider threats by enabling cross-verification between user and LLM for attack prevention. We further consider attack detectability because it forms a defense-in-depth strategy with integrity and source identification.

### 4.2 DESCRIPTION OF THE PROPOSED DEFENSE

**Insight of Defense Design.** Our key idea is to design a defense to ensure the queries from users cannot be manipulated, and are distinguishable from the intermediate prompts from application. In particular, our defense leverages digital signatures (Aki, 1983) to ensure integrity and source identification. By letting users (resp. LLM) sign their respective messages, the LLM (resp. users) could identify the origin of the received messages, and verify whether the message has been ma-

nipulated. Consequently, we can leverage the language model to detect whether the application perturbs the semantics of the intermediate prompt $P_A$ by comparing with the signed user query $P_U$. Utility preservation is achieved by ensuring that the LLM consistently satisfies user queries, while responds to prompts from the application only if no attacks are detected. We note that the LLM lacks capabilities of signing and verifying digital signatures. To address the challenge, we design a new, generic API named Shield, in addition to the API offered by the LLM service provider. Shield is the *first* defense of its kind. It is designed to be lightweight and highly effective. It does not require retraining the LLM and thus is compatible with the SOTA LLM deployments. Figure 5 shows the workflow of Shield.

**Notations.** We define the signature $\sigma$ of a message $m$ as $\sigma = \text{sig}_K(m)$, where $\text{sig}_K$ is a signing algorithm using key $K$. We denote the signed message $m$ as $(m, \sigma)$. The verification of $(m, \sigma)$, denoted as $\text{ver}_K(m, \sigma)$, outputs either true or false. We denote the unique session ID as $id$.

**Overview of our Defense.** Shield mitigates attacks that can occur during both

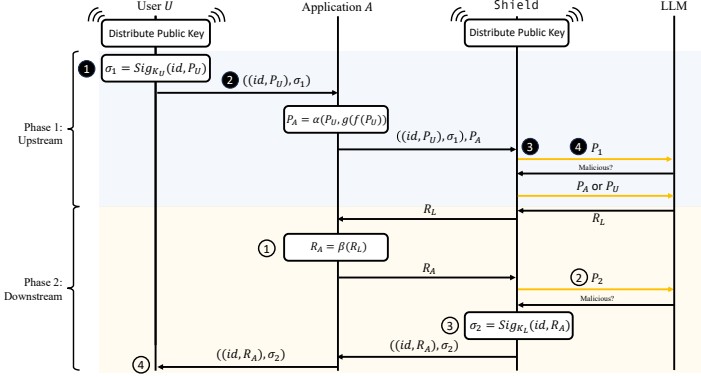

Figure 5: This figure shows the workflow of Shield.

upstream and downstream communications, as shown in Fig. 5. It follows steps ❶ to ❹ for threat mitigation and detection during the upstream communication, detailed below. ❶: The user appends the session ID into its query $P_U$, and signs $(id, P_U)$ using its key $K_U$ as $\sigma_1 = \text{sig}_{K_U}(id, P_U)$. ❷: The signed query is then sent to the application to generate the intermediate prompt $P_A = \alpha(P_U, g(f(P_U)))$. Note that action $\alpha$ cannot tamper with the signed query $\sigma_1$ without compromising the user's signature. ❸: After receiving the intermediate prompt, Shield verifies whether $\text{ver}_{K_U}((id, P_U), \sigma_1)$ holds true. If the result is true, Shield then records the ID and constructs a meta-prompt for LLM to detect attacks as $P_1 = \{$ "System Prompt" : $I_1$, "User" : $P_U$, "Application" : $P_A\}$, where $I_1$ is a system prompt (OpenAI, 2023f) that leverages the instruction-following behavior (Kang et al., 2023) of LLM to guide its response. Prompt template of $I_1$ can be found in Appendix B.3. ❹: Shield then sends $P_1$ to the LLM. If the LLM reports negative on attack detection, the API then transmits $P_A$ to the LLM and requests the response $R_L$, which will further be returned to the application. If the LLM detects attacks, then API only sends the user's query $P_U$ to the LLM for response generation.

Shield follows steps ① to ④ during the downstream communication. ①: After the application receives the response $R_L$ from the LLM, it generates a response $R_A$ and sends it back to Shield. The API then constructs a meta-prompt $P_2 = \{'$System Prompt$' : I_2, '$Core Response$' : R_L, '$Application$' : R_A\}$. System prompt $I_2$ is designed similarly to $I_1$, and is used to detect attacks during the downstream communication. Prompt template of $I_2$ is in Appendix B.3. ②: Shield then sends $P_2$ to the LLM for attack detection. ③: If the LLM detects no attack, then Shield signs $R_A$ as $\sigma_2 = \text{sig}_{K_L}(id, R_A)$, where $K_L$ is the key of Shield. The signed response $((id, R_A), \sigma_2)$ is then returned to the user. If the LLM detects attack, Shield returns $R_L$ to the user with the corresponding signature. ④: After receiving responses from the application, the user executes $\text{ver}_{K_L}((id, R_A), \sigma_2)$. If the verification process returns true, then the user accepts $R_A$ as the response. The workflow of Shield described above is exemplified in Fig. 8 of the appendix, demonstrating how Shield mitigates the toxic risk raised by the outsider threats.

Table 4: Evaluations of attack detectability and utility preservation of Shield against bias and toxic risks. "Neutral" quantifies the percentage of responses that successfully address the users' queries. Other percentage numbers characterize the success rate of Shield in detecting attacks.

| Model | Bias | | | | Toxic | | | |
|---|---|---|---|---|---|---|---|---|
| | Neutral | Pertb-User | Pertb-System | Proxy | Neutral | Outsider-Explicit | Outsider-Implicit | Pertb-System |
| GPT-3.5 | 94% | 100% | 92% | 71% | 100% | 100% | 86% | 100% |
| GPT-4 | 100% | 100% | 100% | 99% | 100% | 100% | 100% | 100% |

### 4.3 Evaluation of Shield

We empirically evaluate the attack detectability and utility preservation of our defense. We quantify the attack detectability by computing the ratio of tests that are correctly labeled as under attack. The utility preservation is evaluated using the Neutral scenario, where there exists no attack. We remark that when attacks are detected, the utility of the LLM-integrated application may degrade. The extent of utility degradation depends on the user query.

We summarize the evaluation results on the online shopping application in Table 4. We first observe that Shield successfully detects the attacks when both GPT-3.5 and GPT-4 are used as LLM services. The latest GPT-4 achieves nearly 100% success rate in detecting attacks across all risks. Furthermore, Shield preserves the utility of LLM-integrated applications. When there exist no attacks (Neutral in Table 4), all responses produced by LLM-integrated applications successfully address the users' queries. We further compare the attack detectability and utility preservation of Shield with a baseline under the toxic risk in Table 13 of the appendix, and show that Shield consistently outperforms the baseline in all settings. Evaluation results against privacy and disinformation risks are in Appendix B.4. We also evaluate Shield in a medical assistance application in Table 15 of the appendix. We prove the integrity and source identification of Shield in Appendix B.2.

## 5 Related Work

**Misuses of LLMs.** The vulnerabilities and risks of LLM have been studied in recent works including (Abid et al., 2021; Bender et al., 2021; Bommasani et al., 2021; Bowman, 2023; Gehman et al., 2020; OpenAI, 2023b; Weidinger et al., 2021; Perez & Ribeiro, 2022; Kang et al., 2023). Indeed, the risks and defects associated with LLMs will be inherited by the downstream applications (Bommasani et al., 2021). In this work, we focus on LLM-integrated applications, which not only inherit the vulnerabilities of LLMs as identified by the aforementioned works, but also open up new attack surfaces due to the presence of untrusted/unverified application developers/vendors. More comprehensive literature review can be found in Appendix C.

**Risk Mitigation Developed for LLMs.** Mitigation strategies against toxic text generation of LLM have been developed. The authors of (Liang et al., 2021) identified the sensitive tokens and mitigated biases by using iterative nullspace projection. Societal bias-aware distillation technique was developed in (Gupta et al., 2022). Compared to (Gupta et al., 2022; Liang et al., 2021) which required tuning or training the model, our approach is lightweight without re-training or modifying the LLM. An alternative approach to mitigate biases of LLMs is to apply filtering-based techniques (Guo et al., 2022; Pavlopoulos et al., 2020; Zellers et al., 2019; OpenAI, 2023d). However, these filtering-based techniques may not be applicable to mitigate our identified vulnerabilities in LLM-integrated applications (see Section 3). More detailed comparison with existing literature is in Appendix C.

## 6 Conclusion and Discussion

In our work, we show that LLM-integrated applications become new attack surfaces that can be exploited by both insider and outsider threat initiators, leading to bias, toxic, privacy, and disinformation risks for users of applications. Our extensive empirical evaluations confirm those risks. To mitigate them, we identify four key properties that defense should satisfy. We design a defense that simultaneously satisfies four properties by providing a new API for the LLM service providers in addition to the LLM-API. Our defense is compatible with any LLMs. Our experimental results demonstrate the efficacy of our defense. We acknowledge that our identified threats can be misused and raise ethical concerns, which is discussed in Section 7.

This paper assumes that both users and LLM are non-malicious. We acknowledge that there may exist additional threats. For example, an adversary may simultaneously exploit the vulnerabilities in the application and LLM to gain extra advantages. Such attacks targeting multiple entities in LLM-integrated applications need to be addressed in future work. In addition, the user may not necessarily be benign and could act maliciously for its own interest. In this paper, we evaluate bias, toxic, privacy, and disinformation risks. We are aware of other potential risks such as discrimination and misinformation (Weidinger et al., 2021). We believe Shield is agnostic to the semantic goals of threat initiator, and is applicable to prevent and mitigate these potential risks.

